# PromoterLCNN: A Light CNN-Based Promoter Prediction and Classification Model

**DOI:** 10.3390/genes13071126

**Published:** 2022-06-23

**Authors:** Daryl Hernández, Nicolás Jara, Mauricio Araya, Roberto E. Durán, Carlos Buil-Aranda

**Affiliations:** 1Department of Electronics Engineering, Universidad Técnica Federico Santa María, Valparaiso 2390123, Chile; daryl.hernandez@usm.cl (D.H.); mauricio.araya@usm.cl (M.A.); 2Laboratorio de Microbiología Molecular y Biotecnología Ambiental, Departamento de Química & Centro de Biotecnología Daniel Alkalay Lowitt, Universidad Técnica Federico Santa María, Valparaiso 2390123, Chile; ro.duran.vargas@gmail.com; 3Department of Informatics Engineering, Universidad Técnica Federico Santa María, Valparaiso 2390123, Chile; cbuil@inf.utfsm.cl

**Keywords:** bacterial promoters, convolutional neural networks, bioinformatics, deep learning, PromoterLCNN

## Abstract

Promoter identification is a fundamental step in understanding bacterial gene regulation mechanisms. However, accurate and fast classification of bacterial promoters continues to be challenging. New methods based on deep convolutional networks have been applied to identify and classify bacterial promoters recognized by sigma (σ) factors and RNA polymerase subunits which increase affinity to specific DNA sequences to modulate transcription and respond to nutritional or environmental changes. This work presents a new multiclass promoter prediction model by using convolutional neural networks (CNNs), denoted as PromoterLCNN, which classifies *Escherichia coli* promoters into subclasses $\sigma^{70}$, $\sigma^{24}$, $\sigma^{32}$, $\sigma^{38}$, $\sigma^{28}$, and $\sigma^{54}$. We present a light, fast, and simple two-stage multiclass CNN architecture for promoter identification and classification. Training and testing were performed on a benchmark dataset, part of RegulonDB. Comparative performance of PromoterLCNN against other CNN-based classifiers using four parameters (Acc, Sn, Sp, MCC) resulted in similar or better performance than those that commonly use cascade architecture, reducing time by approximately 30–90% for training, prediction, and hyperparameter optimization without compromising classification quality.

## 1. Introduction

Bacterial promoters are DNA sequences positioned upstream of the transcription start site (TSS), crucial for recognition by the RNA polymerase (RNAP) [1]. Promoters initiate transcription due to their affinity to the RNAP, and therefore are essential for maintaining cellular homeostasis. Promoter affinity is highly determined by two conserved hexamers in positions −10 and −35 upstream of the TSS [1,2]. The promoter efficiency is also modulated by a spacer region around 17 ± 3 bp length between the hexamers and the adjacent nucleotide sequences (UP element, extended −10 element; (Figure 1A) with lower conservancy in comparison to the −10 and −35 promoter elements [3,4].

The bacterial RNAP consists of six subunits—the core RNAP composed of five subunits ($\alpha_{2}\beta \beta^{\prime }\omega$), and the sigma (σ) subunit or σ factor (Figure 1B). The σ factor reversibly binds to the core RNAP modulating the DNA-binding characteristics of the enzyme, increasing the affinity for promoters [1]. The most abundant sigma factor is $\sigma^{70}$ ($\sigma^{A}$), encoded by *rpoD* and responsible for almost all gene expression. Different σ factors have been identified, each one competing for the core RNAP and initiating the transcription of different genes associated to a specific nutritional status or environmental condition [1,5]. Alternative sigma factors targeting different promoters have been characterized in *E. coli*. These include the $\sigma^{H}$ ($\sigma^{32}$) encoded by *rpoH* involved in heat shock stress response, the $\sigma^{S}$ ($\sigma^{38}$) associated to stationary phase regulation, $\sigma^{E}$ ($\sigma^{24}$) related to extracytoplasmatic functions, $\sigma^{N}$ ($\sigma^{54}$) relevant to nitrogen metabolism, and $\sigma^{F}$ ($\sigma^{28}$) associated to flagellar synthesis and chemotaxis [5]. Each σ factor recognizes and facilitates the binding of RNAP to different promoters with distinct consensus sequences, which hampers the in silico identification of promoter sequences in bacterial genomes [6].

Accurate and fast prediction of promoter sites associated with a σ factor remains a troublesome issue in genomics and molecular biology, despite being highly relevant for gene expression patterns, genetic regulatory networks, and synthetic biology studies, wherein the synthesis of new nucleotide sequences can incorporate undesired promoter sequences [6,7,8]. Traditional methods carried out by low-scale methods (e.g., DNA footprinting, primer extension, electrophoretic mobility shift assay) are slow and time-consuming. Meanwhile, even the use of high-throughput technologies (e.g., RNA-seq, systematic genomic evolution of ligands by exponential enrichment) [9,10,11] still does not compare to the explosive amount of genomic data generated during the last decade [12,13]. In silico prediction based on bioinformatic tools has also been explored based on sequence information, first as position weight-matrices [14,15,16], and later by using machine-learning (ML) techniques. The latter stand out as they do not require manual assembly of the characteristics or patterns to be detected, are free-alignment methods that do not require comparison with known sequences or databases, and work on raw information [6,7,17,18,19,20,21,22]. This feature is particularly relevant because one of the biggest problems identifying promoters is precisely their high variability. Then, depending on the overall context, a sequence may or may not be part of a promoter, as the so-called *Pribnow box* conformed by TATAAT (the −10 promoter element) is typically associated with promoters, but their sole identification does not define them.

To date, there are still few proposals based on neural networks for promoter sequence classification capable of making predictions with specificity and sensitivity values more significant than 80% and even 90%. These techniques generally use data from promoters recognized by $\sigma^{70}$ factor [7,20]. In bacteria, they have been tested on the model strains (*E. coli* str. K-12 substr. MG1655 and *Bacillus subtilis* subsp. *subtilis* str. 168) by using already characterized promoter sequences as positive samples and delivering random sequences of coding sequences (CDS) as the negative sample. These methods generally take the problem as a classification problem, classifying the sequence as either a promoter $\sigma^{70}$ or a non-promoter $\sigma^{70}$ [6,7,20].

In recent years, various strategies have been explored to address the problem described, emphasizing those based on machine-learning techniques. Notably, the following references are based on convolutional neural networks (CNN). Reference [20] proposed CNNProm, consisting of a CNN of a one-dimensional (1D) convolution layer followed by a max-pooling and a fully connected ReLU, with sigmoid-triggered output. Its dataset contains bacterial, human, mouse, and plant sequences, each of 81 nucleotides for bacteria and 251 for the rest. These were coded according to one-hot encoding: A→(1,0,0,0,0), T→(0,1,0,0,0), G→(0,0,1,0), C→(0,0,0,1). For negative examples, random coding sequences were selected. Qian et al., proposed improvements to CNNProm [19]. They used support vector machines to highlight the importance of element sequences of eukaryotic promoters (9 elements included), compressing the non-element sequences of the promoter. The promoter sequences also used one-hot encoding. Subsequently, [7]. proposed DeePromoter, adding a long short-term memory (LSTM) to the architecture. They also stop using coding sequences as negative examples, generating them from the positive ones and replacing random parts, increasing the method robustness against false positives. Another novelty is incorporating dropout layers to increase robustness and prevent overfitting. pcPromoter-CNN [22] presents a convolutional neural network model for promoter prediction and classification of sigma sub-classes following a cascading architecture, performing a binary classification. First, classifying promoters and non-promoters, then for the promoters, it checks if it belongs to $\sigma^{70}$; if not, it continues with $\sigma^{24}$ and so on. Finally, Ref. [18] developed IPromoter-BnCNN, capable of classifying promoters into five sigma categories by using a series of cascading binary classifiers as well as pcPromoters-CNN. In IPromoter-BnCNN, each binary classifier is a CNN of 4 parallel branches.

This study proposes a light two-stage promoter prediction and classification model by using multiclass CNN. We denote this model as PromoterLCNN. The first stage was designed to distinguish between promoters and non-promoters, and the second stage performs the sigma classification by using a multiclass classification model. As stated by previous works in the literature such as [22,23,24], we use Chou et al.’s 2011 [25] five-step rules for a clear presentation and validation of the model. We applied the model to *E. coli* by using RegulonDB v9.3 and 10.7 benchmark databases [26,27], for training and independent testing, respectively. We validate the results by using the K-fold cross-validation technique, examining four performance evaluation metrics to compare them with the best-performing methods found in the literature.

This paper is structured as follows. Section 2 presents our prediction and classification model. Section 3 presents the numerical results and a further discussion. Lastly, Section 4 states several conclusions and final remarks.

## 2. Inputs and Methods

In this section, we present the PromoterLCNN model, following the five-steps rule suggested by [25]. Figure 2 presents an overview of our PromoterLCNN model, and each of the five steps is described in detail as follows.

### 2.1. Benchmark and Test Datasets

Following the path set by previous works [7,18,22], the first essential action for generating a valid statistical promoter predictor is selecting a suitable benchmark dataset for training the model and an independent dataset for testing. We use the same disjointed datasets as in the literature [22,23,24], the benchmark based on experimentally verified promoter sequences and non-promoter ones extracted from coding zones. They are part of RegulonDB version 9.3, in which each entry has 81 bp [26].

The independent test dataset comprises promoter samples only, extracted from RegulonDB version 10.7, which are also experimentally verified [27].

The Training and Test Dataset sample size can be found in Table 1. The entire dataset D is composed of two subsets—the P promoters and the $\bar{P}$ Non-promoter—as stated in Equation (Equation 1):(1)$$D=P\cup \bar{P}.$$

Similarly, the *P* promoter subset contains all sigma sub-classes. Promoter sequences classified as $\sigma^{24}$, $\sigma^{28}$, $\sigma^{32}$, $\sigma^{38}$, $\sigma^{54}$, and $\sigma^{70}$ according to the affinity to each σ factor is.

Accordingly, the *P* promoter subset can be defined as follow in Equation (Equation 2):(2)$$P=\sigma^{24}\cup \sigma^{28}\cup \sigma^{32}\cup \sigma^{38}\cup \sigma^{54}\cup \sigma^{70}.$$

### 2.2. Mathematical Formulation of DNA Sequence

The DNA sequence comprises a large string mixing four nucleotides denoted as A, C, G, and T. Similar to previous works, we use one-hot encoding to transform the DNA sequences to a binary form [20,22]. Each nucleotide is converted to a four-element vector with a single element value equal to one, and all other element values are 0. All numerical representation of nucleotides are as follows:$$\begin{array}{c} A\to (1,0,0,0)\\ C\to (0,1,0,0)\\ G\to (0,0,1,0)\\ T\to (0,0,0,1) \end{array}$$

### 2.3. Model Architecture

The prediction and classification model denoted as PromoterLCNN is composed of two stages. The first stage recognises the DNA sequence as a promoter or not a promoter. Next, the second stage serves as a multiclass classification layer for the acknowledged promoters of the previous stage, identifying the sigma subclass associated with the DNA sequence. Both stages are CNN architectures. We trained the first stage of the model by using the promoter and non-promoter benchmark database (D), and for the second stage, we trained the multiclass model by using all sigma promoters benchmark database (P) displayed in Table 1.

Next, further details of each stage are explained.

#### 2.3.1. First Stage

As previously mentioned, the first stage focuses on the binary recognition of promoters and non-promoters. To face this endeavour, we design an architecture based on convolutional neural networks, illustrated in Figure 3.

This stage of the architecture consists of 2 single-dimensional convolution layers in tandem, followed by batch normalisation, max-pooling, and dropout, preventing over-fitting. Later, the features obtained by the model are flattened on a flatten layer. Finally, the set passes through two fully connected dense layers for the binary classification.

#### 2.3.2. Second Stage

Thereupon, the second stage performs a multiclass classification of promoter sub-classes. This stage of the architecture uses the same first-stage convolutional neural network architecture with the sole difference that the possible classification exits are six instead of two. This stage of the architecture is displayed in Figure 4.

Therefore, this stage consists of two single-dimensional convolution layers. Next, batch normalisation, max-pooling, dropout, flatten, and finally, two fully connected dense layers are used for the multiclass classification of sigma sub-classes.

A hyperparameter tuning is performed to select the most acceptable parameters on all the layers in both stages’ architectures. Next, we explain this tuning process.

#### 2.3.3. Hyperparameter Tuning

For choosing the finest parameters for the two convolutions, pooling, dropout, and the last two dense layers, hyperparameter tuning is performed. Table 2 presents the candidate values of hyperparameters used for the hyperparameter optimization process.

The hyperparameter tuning process was performed by Keras Tuner by using the Hyperband method. This process is time-consuming; thus, some selected hyperparameters were considered, as shown in Table 2.

In both stages, the first convolution layer uses 128 filters of kernel size 5 with $10^{-3}$ L2 kernel regularizer and $10^{-2}$ bias regularizer. The second convolutional layer uses 128 filters of kernel size 9, with $10^{-3}$ and $10^{-5}$ for the L2 kernel regularizer and bias regularizer, respectively. Both convolution layers use a ReLU activation function. The max-pooling layer uses a pool size of 2 and 2 strides. Then the dropout layer drops 45% of the features previously obtained. Finally, the two fully connected dense layers have 32 neurons using a ReLU activation function, with L2 kernel regularizer and bias regularizer values of $10^{-3}$.

### 2.4. Performance Metrics

We use a straightforward strategy for measuring our model performance. We use the same strategy used by several works in the literature [18,22]. The quality of our prediction and classification model is defined by four metrics, Acc, Sn, Sp, and MCC, corresponding to the accuracy, sensitivity, specificity, and Matthews correlation coefficient, respectively. Their definitions are as follows:(3)$$Acc=\frac{TP+TN}{TP+TN+FP+FN^{’}}$$
(4)$$Sn=\frac{TP}{TP+FN^{’}}$$
(5)$$Sp=\frac{TN}{TN+FP^{’}}$$
(6)$$MCC=\frac{TP\times TN-FP\times FN}{\sqrt{\left(TP+FP)(TP+FN)(TN+FP)(TN+FN\right)}}$$
in which TP, TN, FP, and FN stand for the true positives, true negatives, false positives, and false negatives, respectively.

## 3. Results and Discussion

We train our model by using k-fold validation with a k value equal to 5. The performance results are summarized in Figure 5 and Figure 6, illustrating the results for our PromoterLCNN (Lc), and obtaining performances better than or similar to pcPromoter-CNN (Pc) and iPromoter-BcNN (Bc) methods, measuring their accuracy (Acc), sensitivity (Sn), specificity (Sp) and Matthews correlation coefficient (MCC), for the training dataset (Figure 5) and the independent test dataset (Figure 6) mentioned in Section 2.1. We remark that these figures are also presented as a heatmap, in which the colour intensity indicates the quality of each value displayed here.

As displayed in Figure 5, the results using the training database show that our approach (Lc) performs better than the pcPromoter-CNN (Pc) method for every metric and every promoter. In fact, a weighted average of accuracy with respect to the number of elements in each class gives 96.7% for Lc and 91.2% for Pc. From the sensitivity results, our method tends to produce a few more misclassifications than iPromoter-BcNN for this dataset, and the specificity row suggests that these corresponded to false-negative non-promoters as the dataset is balanced. In other words, the approach is very efficient detecting and classifying promoters, with minor contamination of non-promoters. In any case, the sensitivity values of the PromoterLCNN drop or improve a few points concerning iPromoter-BcNN, but it does not significantly fail like the pcPromoter-CNN does for $\sigma^{38}$ and $\sigma^{32}$. The MCC shows that the confusion matrix quality for most classes is very similar for the two leading methods. A few misclassifications on the training data might indicate that the network is generalizing correctly, as opposed to overfitting problems found in shallow or classical learning methods. To verify this, we must achieve similar performance in the testing dataset.

Regarding the results over the test dataset presented in Figure 6, our approach outperforms the pcPromoter-CNN, as the weighted averages are 89.6% for Lc and 83.0% for Pc, and achieves comparable (and even slightly better) results than iPromoter-BcNN in terms of accuracy. PromoterLCNN produces similar or even fewer false negatives, as illustrated by the specificity values of promoters (Sp). However, it is important to recall Table 1 for a tempered analysis: each promoter class has only a few examples, and there are no non-promoters. Therefore, the most relevant statistics here are those obtained for $\sigma^{70}$ and $\sigma^{24}$ (i.e., n≥30). These statistics show that our approach is better at classifying promoters, at the cost of non-detecting a few of them (false negatives) compared to iPromoter-BcNN.

Establishing the competitive performance of our approach is vital to notice the parsimony in our proposal. We present an architecture appreciably lighter than pcPromoter-CNN and iPromoter-BcNN. The first one uses a cascading architecture for classifying first promoters and non-promoters, and later sigma sub-classes sequentially, with nine different layers each [22]. The second one uses the same cascading architecture, with four parallel layers (4 layers each branch), converging into 4 layers at the end [18]. On the contrary, our method has only two stages with eight different layers each. This feature significantly diminishes the computing time for training, prediction, and hyperparameter optimization processes, taking a tenth of the time compared to iPromoter-BcNN, and 30% less than pcPromoter-CNN.

## 4. Conclusions

This work presents a two-stage promoter prediction and classification model by using a multiclass convolution neural network called PromoterLCNN. The first stage of the architecture attempts to recognize between promoters and non-promoters, and the second stage engages the sigma classification by using a straightforward multiclass classification model, in contrast to standard approaches found in the literature. We use Chou et al.’s five-step rules for our model presentation and validation process by using *E. coli* databases found in RegulonDB v9.3 and v10.7 benchmark databases for training and independent testing, respectively.

We used a K-fold cross-validation training and assessed the results with an independent test dataset. We found out that our method outperforms one of the most competitive methods in the literature. For the more recent state-of-the-art approaches, our proposal has competitive results in accuracy, with better promoter-type classification. Remarkably, PromoterLCNN has a lighter architecture than other models, leading to a shorter time for the hyperparameter tuning, training, and prediction processes without compromising classification quality, an attractive quality for molecular or synthetic biologists working with nucleotide sequences on a daily basis. By using part of a genome or a newly synthetized sequence as input data, PromoterLCNN might help researchers and users working in the field of bacterial genomics, molecular biology, and bioinformatics to identify bacterial promoters and classify them into each of the σ subclasses validated in this study.

## Figures and Tables

**Figure 1 genes-13-01126-f001:**
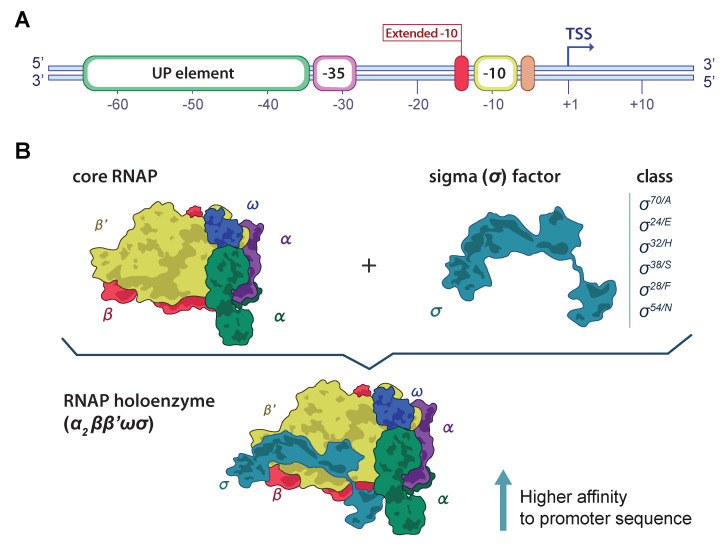
Graphic: representation of the bacterial DNA promoter and the RNA polymerase (RNAP) holoenzyme. (**A**) Key elements of the bacterial DNA promoter are crucial for RNAP affinity and binding. The coloured rounded rectangles represent key elements of the bacterial promoter. These include the UP element, the −35 and −10 hexamers, the extended −10 element, and the discriminator sequence. The transcriptional start site (TSS) is represented at the +1 position. (**B**) The RNAP holoenzyme components and subunits. The two components of the functional RNAP complex are illustrated in the upper part: the core RNAP enzyme, composed of the α, β, and ω subunits; and the sigma (σ) factor, which improves the RNAP–DNA affinity for transcription. At the bottom, the binding of the σ factor to the core enzyme completes the RNAP holoenzyme ($\alpha_{2}\beta \beta^{\prime }\omega \sigma$).

**Figure 2 genes-13-01126-f002:**
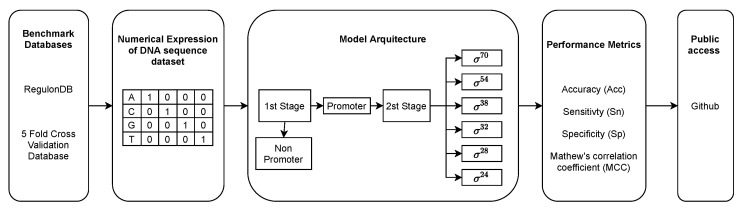
Overview of the PromoterLCNN model.

**Figure 3 genes-13-01126-f003:**
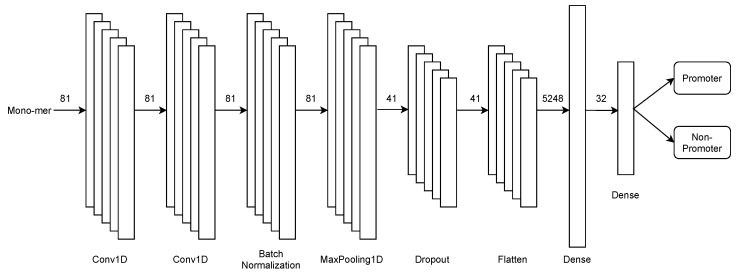
First-stage multiclass CNN architecture for promoters recognition in PromoterLCNN.

**Figure 4 genes-13-01126-f004:**
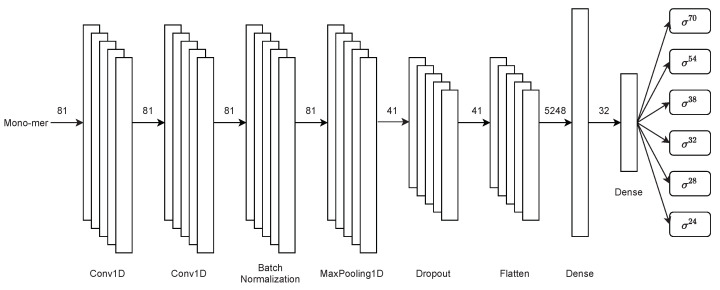
Second-stage multiclass CNN architecture for sigma classification in PromoterLCNN.

**Figure 5 genes-13-01126-f005:**
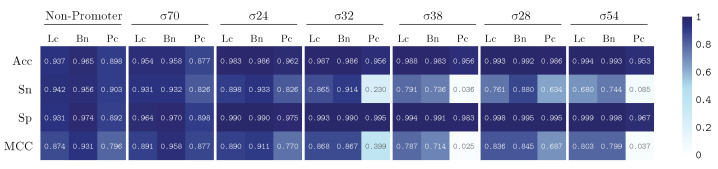
Performance over the training dataset for PromoterLCNN (Lc), pcPromoter-CNN (Pc) and iPromoter-BcNN (Bc), measuring accuracy (Acc), sensitivity (Sn), specificity (Sp) and Matthews correlation coefficient (MCC).

**Figure 6 genes-13-01126-f006:**
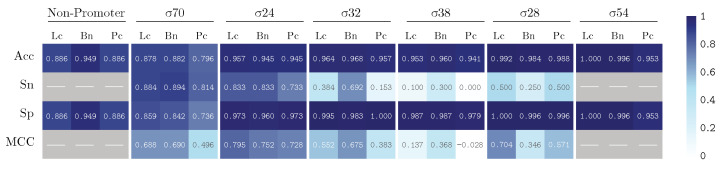
Performance on the test dataset for PromoterLCNN (Lc), pcPromoter-CNN (Pc) and iPromoter-BcNN (Bc), measuring accuracy (Acc), sensitivity (Sn), specificity (Sp) and Matthews correlation coefficient (MCC).

**Table 1 genes-13-01126-t001:** Training and test dataset sample sizes.

Classes	Training Dataset	Test Dataset
Promoter (P)	2860	256
Non-Promoter ($\bar{P}$)	2860	0
$\sigma^{70}$-Promoter	1694	199
$\sigma^{24}$-Promoter	484	30
$\sigma^{32}$-Promoter	291	13
$\sigma^{38}$-Promoter	163	10
$\sigma^{28}$-Promoter	134	4
$\sigma^{54}$-Promoter	94	0

**Table 2 genes-13-01126-t002:** Hyperparameter tuning parameters and their candidate values.

Hyper Parameter	Candidate Values
Number of Convolution Filters	16, 32, 64, 128
Convolution Kernel Size	3, 5, 7, 9
Kernel Regularizers (L2)	1× 10${}^{-5}$, 1 × 10${}^{-4}$, 1 × 10${}^{-3}$, 1 × 10${}^{-2}$
Dropout Rate	0.15, 0.20, …, 0.50
Dense Layer Neurons	8, 16, …, 64

## Data Availability

PromoterLCNN is available in the following github repository: https://github.com/occasumlux/Promoters (accessed on 1 June 2022).

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
