# Peer review of "PromoterLCNN: A Light CNN-Based Promoter Prediction and Classification Model"

_genes, 2022, doi:10.3390/genes13071126_

Round 1
Reviewer 1 Report
Dear authors of "PromoterLCNN: A Light CNN-based Promoter Prediction and Classification model", your work would have a great contribution to bacterial genetics in terms of improving promotor prediction. Your PromoterLCNN looks very promising.
Here are some suggestions/ comments:
- - Line 18-27: I suggest to add a picture showing the structure of promoters and RNAP subunits which will support the information in the text.
- Table 1: the numbers in the training and the tested dataset varies hugely for each class. Would such variation has an effect on the accuracy and sensitivity of the predication?
- - Table 3 and Table 4: is it possible to show this data on graphs to make it easer to compare between the three models?
- - Lines 166-167: "our approach 166 (Lc) outperforms the pcPromoter-CNN (Pc) method for every promoter class"; for some of them, the difference between Lc and Pc looks less than 2% (e. g. σ28, σ32, σ38, and σ70). How would you justify that the difference in accuracy is huge for some promoters and very miner for others? would you consider less that 2% as significant difference? - add more explanation.
Thank you,
.
-
-
Author Response
Dear Reviewer
Thank you for allowing a resubmission of our manuscript, with an opportunity to address your comments. We are uploading our point-by-point response to the comments. Please, see the attachment.
Sincerely,
The authors

Reviewer 2 Report
Hernàndez and coworkers present the manuscript “ PromoterLCNN: A Light CNN-based Promoter Prediction and Classification model“. Data is clear and conclusions are supported by the good performance of the model.
My only suggestion it’s to include specific examples of prediction for known promoters in a friendly format for non-bioinformatics experts, as finally these models will be tested and used by biologists. If not the paper remains too technical.
Author Response
Dear Reviewer
Thank you for allowing a resubmission of our manuscript, with an opportunity to address the comments. We are uploading our point-by-point response to the comments. Please see the attachment.
Sincerely,
The authors
